# Nonclassical Attack on a Quantum Key Distribution System

**DOI:** 10.3390/e23050509

**Published:** 2021-04-23

**Authors:** Anton Pljonkin, Dmitry Petrov, Lilia Sabantina, Kamila Dakhkilgova

**Affiliations:** 1Institute of Computer Technology and Information Security, Southern Federal University, 347900 Taganrog, Russia; dapetrov@sfedu.ru; 2Junior Research Group Nanomaterials, Faculty of Engineering and Mathematics, Bielefeld University of Applied Sciences, 33619 Bielefeld, Germany; lilia.sabantina@fh-bielefeld.de; 3Faculty of Information Technology, Chechen State University, 364024 Grozny, Chechen Republic, Russia; puma-i@mail.ru

**Keywords:** quantum key distribution, single-photon mode, synchronization, algorithm, detection probability, vulnerability

## Abstract

The article is focused on research of an attack on the quantum key distribution system and proposes a countermeasure method. Particularly noteworthy is that this is not a classic attack on a quantum protocol. We describe an attack on the process of calibration. Results of the research show that quantum key distribution systems have vulnerabilities not only in the protocols, but also in other vital system components. The described type of attack does not affect the cryptographic strength of the received keys and does not point to the vulnerability of the quantum key distribution protocol. We also propose a method for autocompensating optical communication system development, which protects synchronization from unauthorized access. The proposed method is based on the use of sync pulses attenuated to a photon level in the process of detecting a time interval with a signal. The paper presents the results of experimental studies that show the discrepancies between the theoretical and real parameters of the system. The obtained data allow the length of the quantum channel to be calculated with high accuracy.

## 1. Introduction

This research was inspired by the works “Quantum man-in-the-middle attack on the calibration process of quantum key distribution” [1] and “Device calibration impacts security of quantum key distribution” [2], which describe attacks on the calibration system. In the beginning, it is necessary to clarify several important nuances about our research: the experiments were carried out with a two-pass quantum key distribution system (QKDS) Clavis^2^; we do not examine the security of the quantum BB84 protocol and do not claim that our attack is an attack on the BB84 protocol; and we do not test the strength of quantum keys and do not claim that the described attack affects the strength of the keys. These are important notes for understanding the aims of the paper. The quantum key distribution process and the synchronization process are different. There are many articles in the literature that describe these processes in detail. There are attacks on both quantum protocols and the synchronization process, but there is practically no literature describing attacks on the synchronization process. Our experiment was carried on the real Clavis^2^ quantum key distribution system. These are two stations connected by a quantum channel-optical fiber. In real operating conditions, QKDS have many loopholes for an attacker. This is not about quantum cryptography protocols that are reasonably secure. We are referring to the technical imperfection of systems. The authors [1,2] discuss such imperfections and show that an attacker can use them for attacks. It is important to understand that the purpose of an attack on the QKDS may not only be the acquisition of a secret key. Implementation of a controlled interference can also be a target of an attacker. From the user’s point of view, this looks like a technical failure of the system, and there are two options: the user understands that the failure was caused by an attacker, or the user does not detect the attacker. In this work, we will show experimentally how it is possible to interfere with the normal operation of the QKDS without revealing itself.

The basic principles of quantum cryptography are absolute theoretical secrecy of the transmitted data and the impossibility of unauthorized access to it. For cryptographic systems, the security issue is formulated as the problem of distributing the encryption key between legitimate users. Quantum cryptography systems solve the problem of generating and distributing the encryption key using methods that are based on the laws of quantum physics and are implemented in quantum key distribution systems. In the description of quantum key distribution systems, much attention is paid to the operation of quantum protocols. The main problem is the insufficient study of the synchronization process of quantum key distribution systems. This paper contains a general description of quantum cryptography principles. A two-way plug and play fiber-optic quantum key distribution system with phase coding of photon states in synchronization mode was examined. A quantum key distribution system was built on the basis of the scheme with automatic compensation of polarization mode distortions. Single-photon avalanche diodes were used as optical radiation detecting devices. The operation of such systems is impossible without the process of station coordination, i.e., synchronization of the transmitter and receiver separated in space. In the QKDS, synchronization consists of a high-precision determination of the length of the optical pulse propagation path and is based on the registration of the moment when the synchronizing pulse is received by photodetectors.

## 2. Experiment and Simulation

### 2.1. Signal Level in the QKD System

The most appropriate form of synchronization signal for the QKDS is a periodic sequence of optical pulses [3]. In this case, the time markers are the pulses themselves, and the measurement process consists of dividing the entire follow-up period into time intervals. The conversion of a photon to a primary electron is registered in each time interval. The results of live tests of a quantum cryptographic network based on the IDQuantique Clavis^2^ 3110 QKD system are described in [4,5,6,7], and it is shown that the synchronization process generates multiphoton pulses, and the photodetectors operate in linear mode. Using the constructed energy model of the current Clavis^2^ 3110 QKD system, we show that the synchronization mode does not involve algorithms for controlling the emission power. Figure 1 shows the dependence of the number of photons in the pulse on the length of the quantum channel. The quantum channel is a fiber-optic communication line connecting two stations of the QKD system. Dependencies demonstrate three synchronization modes and take into account the following complex losses: in the optical fiber at the junction points, and total losses in the encoding station (−47.7 dBm). The energy model of the QKD system describes the characteristics of the detection equipment. In the process of high-precision determination of the length of a quantum channel, pulses are sent from the transmitting station to the encoding station, where they are reflected from the Faraday mirror and follow back along the same optical path. The process is divided into three stages, for each of which the pulse power values correspond to P1 = −48.3 dBm, P2 = −55.8 dBm, and P3 = −24.2 dBm. The values of P1, P2, and P3 were obtained experimentally using Yokogawa AQ2202 equipment. The photon energy with the refraction index for the Corning^®^SMF-28e+ fiber is equal to
(1)$$E\left(p\right)=\frac{h\frac{c}{n}}{\gamma }=\frac{6.62\cdot 10^{-34}\cdot 2.01\cdot 10^{8}}{1550\cdot 10^{-9}}=0.0085\cdot 10^{-17}$$

Repetition rate f1 = 800 Hz, f2 = 800 Hz, f3 = 5 MHz, and pulse duration τ = 1 ns. The pulse duration is the same for the three modes. We performed the simulation based on the equation. The graphs were plotted using the classical formula for expressing the number of photons in terms of the pulse energy at a known repetition rate, taking into account the refraction index of the emission in the fiber.

The dependences clearly demonstrate that only when the quantum channel is L = 50 km long (taking into account the resulting losses and the double path of movement of the pulses), the average number of photons in the pulse approximates to unity (the average value of the three synchronization stages). The ordinate axis shows the resulting value, i.e., the pulse with this number of photons passed the distance L × 2 and entered the photodetector. It is apparent that the first stage had the most powerful energy characteristics. The latter was related to the need to ensure the highest probability of detecting the reflected signal at the first stage, since an erroneous detection or omission of the signal at the first stage will cause a complex detection error at subsequent stages. Note that the power of optical synchronizing pulses is constant for all values of the length of the quantum channel, i.e., the system does not adjust the laser power depending on the length of the quantum channel. A pulse with the number of photons m >> 10 is called a multiphoton pulse, 1 < m < 10 is a photon pulse, and m < 1 is a single-photon pulse. Therein, a single-photon should not be perceived as a division of a photon, but as the presence of a signal in each j-th pulse.

We showed experimentally that the multiphoton mode of calibration in the quantum key distribution system is a vulnerability. Note that the purpose of unauthorized access may be not only to intercept and read information, but also to synchronize the attacker’s equipment in order to interfere with the work of the QKDS [8,9,10].

### 2.2. Experimental Attack on a Quantum Channel and Analysis

We configured the experimental design (Figure 2), where the quantum communication system stations were located in adjoining rooms. A quantum channel of variable length was organized between them. Corning^®^SMF-28e+ optical fiber coils with lengths (L) of 1, 2, 4, and 25 km were used for this. At the junction points of the optical coils, two fiber-optic couplers with division coefficients were connected in series: kC1 (70%, 30%) and kC2 (90%, 10%). The output of the transmitting station was connected to the input of the divider kC1, and the output of the divider kC1 (70%) was connected to the output of the divider kC2 (90%). The input of the kC2 divider was connected to the quantum channel in the direction of the receiver station. Outputs kC2 (10%) and kC1 (30%) were connected to an optical power meter (Yokogawa AQ2202) to capture signals.

Note that the implementation of couplers in the optical communication channel was not technically difficult. The latter was provided by two welded joints in the fiber-optic communication line. The presence of two couplers allows one to calculate the time of re-reflection, since the moment of interception of an optical pulse in only one direction does not give complete information to the attacker about the operation of the system. It is crucial to intercept the optical pulse during the reverse propagation of the reflected signal. With information about the re-reflection time, an attacker can calculate the exact distance to the recipient’s station and back [11,12,13,14,15,16]. This data allows one to perform some attacks on quantum communication protocols, for example, an attack in which the operation of the coding station is simulated. The attacker inserts their equipment instead of the encoding station and sends substitution signals to the transmitting station’s photodetectors at the right time. The aim of our experiment was to prove the possibility of successful implementation of an attack on a quantum communication system by interference with the calibration stage.

In the described design, the QKD system is put into operation mode. The synchronization process and the operation of the quantum protocol BB84 function normally without critical errors, i.e., the presence of two power couplers in the optical communication channel is not detected by the system and does not affect its operation. Keys are formed in cycles, and the synchronization processes successfully. In this mode, the experiment lasted 24 h, and the system functioned without failures. After the signals at outputs kC1 (30%) and kC2 (10%) were repeatedly recorded, we connected the optical emission source (Yokogawa AQ2202) to the output kC2 (10%). The connection of the emission source also did not affect the operation of the QKDS. Further, at random times, we provided a signal-interference (τ = 1 ns, f = 270 Hz) to the output of the coupler kC2 (10%). The duration of interference activation varied from 5 s to 10 min. In interference mode, the system did not stop operating and did not issue errors but initiated the synchronization process again. After synchronization, the quantum protocol operation was restored, and the key distribution process resumed. We performed a simulation. We clearly demonstrated the effect of interference on the operation of the quantum key distribution protocol. Figure 3 shows the dynamics of the measured quantum error (QBER).

We can see that the graph does not contain any critical changes. Analysis of the dynamics of quantum error does not allow for the detection of unauthorized interference in the operation of the system. The latter is also confirmed by the graph in Figure 4, which shows the dynamics of generated quantum keys.

Figure 4 shows the number of keys that are cyclically accumulated in the buffer. Note that the length of a single key is 512 bits. The dependencies in Figure 3 and Figure 4 are presented for the length of the quantum channel L = 25,732 m. The graph in Figure 4 also does not indicate when the system was affected by the interference. If we consider the approximation of this dependence on the time axis, the time delay with an error of about 10% of the average key generation cycle will be visible in the intervals with interference enabled. This delay occurs periodically during the operation of the QKD system and may be due to the presence of in homogeneities in the quantum channel or physical changes in the optical fiber due to temperature influences. Thus, the time dependence analysis also does not allow for the detection of the presence of couplers in the communication channel or indicate unauthorized interference. Let us turn to Figure 5 and Figure 6. Figure 5 shows statistics of accumulated quantum keys and QBER at different optical link lengths without using couplers (i.e., without introducing interference).

The graphs show that the maximum number of accumulated keys for 6 iterations is 9546, with a quantum channel length (L) of 7880 m. The graph shows a significant difference when the length of the fiber optic cable is 50,456 m. Here, the number of keys generated in one iteration differs significantly from the same value for a shorter length of the fiber optic link, while the growth dynamics is preserved. This dependence behavior is due to the fact that the limit length of the quantum channel introduces significant attenuation in the signal. The values 8.76 < QBER < 9.54 for a quantum channel length of 50,456 m are also high, but these values are not critical, because they do not exceed the calculated value QBER = 11%. Comparing the dynamics of changes in the number of accumulated keys and QBER in the presence of couplers and without them, let us turn to the dependencies in Figure 3, Figure 4, Figure 5 and Figure 6 that are plotted for the length of the quantum channel in 25,732 m. The QBER value is within 2.3 < QBER < 3.1 if there are couplers, and within 2.8 < QBER < 5.7 if there are no couplers. These values are valid and do not indicate the presence of an attacker in the communication channel. Moreover, in the experiment, the values in the absence of couplers exceeded the values in the presence of couplers. The latter indicates that external destabilizing factors have a more significant impact on QBER than the presence of additional prepared connections in the communication channel.

When looking at graphs that reflect the accumulated keys, it is clear that for six iterations, the values do not differ significantly on the two curves (the average number of 512-bit keys per iteration is about 300). Analysis of the results confirms the conclusion that the presence of couplers in the communication channel and the impact of interference do not affect the statistical data of the quantum protocol. A similar conclusion can be drawn when considering the approximated curve on a time chart.

## 3. Single-Photon Synchronization Method

The results of the experiment show the vulnerability of the synchronization process QKDS and prove the possibility of interfering with the system, while remaining unnoticed. Note that the classical method of controlling the emission power in a quantum communication channel does not allow for detection of the presence of couplers. Under ideal experimental conditions, when the quantum channel consists of a continuous fiber (coil), the couplers can be detected using a reflectometer. In this case, it was possible to see attenuation of 0.2–0.4 dB at the places of split joints. If only welded joints are used, the presence of losses is almost impossible to detect. In real conditions, the completed length of the quantum channel does not exceed 1 km, and the presence of fiber optic splice closure is an integral part of the communication system. Fiber optic splice closure and inhomogeneities of optical fiber introduce additional attenuation and hide the possible presence of unauthorized connection to the communication channel. The reflectometric detection method does not allow one to distinguish legitimate inhomogeneities (of different types) from illegitimate ones.

We should also mention the quantum effects of the environment [17,18]. Note that the quantum fluctuations are not described by classical functions and cannot be compensated. Moreover, such quantum effects could be influencing the system, but it is expected that their effects would be small. Of course, such effects must be taken into account, and their influence on the quantum system should be investigated. There are environmental effects that can affect the physical properties of the fiber. For example, temperature tends to change the physical length of a fiber under certain conditions, but it is compensated for by checking the length in the program.

We propose a method that provides protection against an attack on the QKDS during the synchronization process. A distinctive feature of the method is the use of synchronization pulses weakened to a single-photon level. In this case, the optical signal is attenuated at the encoding station by a controlled attenuator, and the value of the insertion loss is calculated so that after reflection from the Faraday mirror, the average number of photons (m) in the synchronizing pulse is 0.1–0.5. Registration of single-photon pulses is performed by avalanche photodiodes in Geiger mode.

The maximum length of the fiber optic link in QKDS is L = 100 km. Taking into account the back propagation of emission to avoid overlapping of back transmitted pulses at L = 100 km, the repetition period is $T_{s}=2\times L/v_{fiber}$ ≈ 1 ms. Therefore, the maximum repetition rate of optical pulses should not exceed $f_{s.max}=1/T_{s}\approx$ 1 kHz. The repetition period $T_{s}$ is divided into $N_{w}$ time intervals with duration $\tau_{w}$ in such a way that $T_{s}=N_{w}\times \tau_{w}$. All intervals are analyzed sequentially. Each interval is analyzed N times, where N is the selection size. The pulse duration $\tau_{s}=1\;$ ns and $\tau_{w}=(2\ldots 4)\times \tau_{s}$. Absolute stability of the repetition period $\Delta T_{s}$ and the duration $\Delta \tau_{s}$ is assumed. In each interval, the number of accepted photoelectrons and/or dark current pulses (DCP) are recorded. After polling all $N_{w}$ time intervals, an array of values is generated as follows:$$\left\{n_{w.N}\left(j\right),\;j=\bar{1,N_{w}}\right\}=\left\{n_{w.N}\left(1\right),\;n_{w.N}\left(2\right),\ldots ,n_{w.N}\left(j\right),\ldots ,n_{w.N}\left(N_{w}\right)\right\}$$

At the values of $\tau_{s}$ and $\tau_{w}$, the synchronizing pulse can lie entirely within one time interval or lie on the border of two neighboring ones. In the first case, the values $n_{w.N}\left(2\right),\;\ldots ,\;n_{w.N}\left(j\right),\;\ldots ,\;n_{w.N}\left(N_{w}\right)$ in $N_{w}-1$ intervals are described by Poisson’s law with the parameter ${\bar{n}}_{d.N}=N\times \xi_{d}\times \tau_{w}$. At the same time, in the interval with a synchronizing pulse, the number $n_{w.N}\left(1\right)$, with the parameter ${\bar{n}}_{w.N}=N\times \xi_{d}\times \tau_{w}+N\times {\bar{n}}_{s}$. Here $\xi_{d}$ is the rate of occurrence of DCP, ${\bar{n}}_{s}$ is the average number of the photoelectrons registered for the duration of the pulse.

If the pulse lies in two neighboring intervals, then random values $n_{w.N}\left(3\right),\;\ldots ,\;n_{w.N}\left(j\right),\;\ldots ,\;n_{w.N}\left(N_{w}\right)$ in $N_{w}-2$ noise intervals are described by Poisson’s law with the parameter ${\bar{n}}_{d.N}=N\times \xi_{d}\times \tau_{w}$, and in neighboring intervals are the numbers $n_{w.N}\left(1\right)$ and $n_{w.N}\left(2\right)$, respectively, with parameters ${\bar{n}}_{w1.N}=N\times \xi_{d}\times \tau_{w}+N\times {\bar{n}}_{s1}$ and ${\bar{n}}_{w2.N}=N\times \xi_{d}\times \tau_{w}+N\times {\bar{n}}_{s2}$. Here ${\bar{n}}_{s1}={\bar{n}}_{s}\times \left(1-\tau_{w}/t_{1}\right)$ and ${\bar{n}}_{s2}={\bar{n}}_{s}-{\bar{n}}_{s1}$ are, respectively, the average number of photons registered in neighboring intervals with the condition that the moment of occurrence of single-photon pulse $\left(t_{1}\right)$ belongs to the first interval. Noise intervals should be understood as analyzed intervals in which the signal is not recorded. In such intervals, noise values can be recorded—the DCP of the photodetector [12,13]. To analyze the process of detecting a synchronizing signal using single-photon pulses, the laws of probability of the distribution density are applied [14].

The analytical expression (2) is used for calculating the probability of correct detection of the signaling interval ($P_{D}$).
(2)$$P_{D}=\overset{\infty }{\underset{n_{w.N}=1}{\sum }}\frac{{\left(\bar{n_{w.N}}\right)}^{n_{w.N}}}{n_{w.N}!}\cdot exp\left[-\bar{n_{w.N}}\right]\cdot P_{d.N}\left\{n_{w.N}\right\}$$

Here
(3)$$P_{d.N}\left\{n_{w.N}\right\}={\left(\overset{n_{w.N}-1}{\underset{n_{d.N}=0}{\sum }}\frac{{\bar{n_{d.N}}}^{n_{d.N}}}{n_{d.N}!}\cdot exp\left(-\bar{n_{d.N}}\right)\right)}^{N_{w}-1}$$
represents the probability of registering no more than ($n_{w.N}-1$) DCP in all ($N_{w}-1$) noise time intervals during the analysis, provided that $n_{w.N}$ photoelectrons and DCP are registered in the signal time interval for a selection of size N. Taking into account the value $N_{w}$, the average number of DCP per sample in the noise interval tends to zero. This allows summation in the formula only for 2 values of $n_{d.N}$ equal to 0 and 1. Simplifying expression (2), we get
(4)$$P_{D}=exp\left(-N_{w}\cdot \bar{n_{d.N}}+\bar{n_{d.N}}\right)\bar{n_{w.N}}\cdot exp\left(-\bar{n_{w.N}}\right)\;+\left[1-exp\left(-\bar{n_{w.N}}\right)-\bar{n_{w.N}}\cdot exp\left(-\bar{n_{w.N}}\right)\right]\cdot {\left(1+\bar{n_{d.N}}\right)}^{N_{w}-1}.$$

The simulation results show that the divergence of the calculation results for Equations (2)−(4) do not exceed 0.02% over the entire variation range in the number of time intervals. The registration validity condition for no more than one photoelectron and/or DCP is typical for a single-photon avalanche photodiode. This proves that it is possible to use expression (4) to calculate the probability of correctly detecting the time interval during the synchronization of the QKDS, provided that $\bar{n_{w.N}}\ll 1$. An important parameter of the avalanche photodiode is the recovery time of the operating mode ($\tau_{dead}$). In the proposed method, the time interval poll is performed sequentially in each frame, i.e., one-time interval is analyzed for the repetition period (T); here $T\gg \tau_{dead}$. This approach allows the recovery time of the working mode of the photodetector to be ignored when calculating. Another distinctive feature of the single-photon mode of operation of the photodetector is the quantum efficiency coefficient of the photocathode (k), which must be taken into account when simulating. Let us look at the graphs in Figure 7, which demonstrate the dependence of the probability of correctly detecting the time interval with signal on the selection size. Dependencies are plotted using Equation (4). The developed method involves the use of a weakened optical synchronizing pulse with an average number of photons 0.1 < m < 1. Thus, given the critical values of the average number of photons per pulse, the frequency of DCP and the quantum efficiency of the photocathode, the variable value is only the selection size in each time interval. Let us explain that the DCP of the photodetector are its shot-noise, which can cause an avalanche effect [15,16,17].

The graphs show that the probability of correct detection reaches maximum values $\left(P_{D}>99.3\%\right)$ already at the selection size N = 30 (without taking into account quantum efficiency) and at N = 150 with taking into account quantum efficiency. Note that the typical selection size of the current Clavis^2^ 3110 system is 800. Next, let us consider the simulation results that show the influence of the frequency of DCP and the selection size on the probabilistic characteristics of detecting the signaling time interval. The task of simulation is to find the optimal values of N and DCP, at which the maximum probability of detection is achieved. Calculations were made taking into account the above average quantum efficiency of the photocathode (k = 25%). Figure 8 shows the results of simulation of the algorithm for detecting a single-photon signal. The graphs show the dependence of the probability of correct detection of the signaling interval on selection size for different values of DCP.

The average amount of photoelectrons (m) in a pulse is 0.1. The graph shows that at the minimum values of the selection size (128 < N < 32), the probability of detection ($P_{D}$) is no more than 80%, and the number of DCP does not matter. This behavior of the curves is explained by a small difference in the number of DCP and photoelectrons in time intervals. The divergence is leveled when the selection size increases. On the other hand, if the value of DCP > 200, the selection size does not matter, since the probability of detection ($P_{D}$) over the entire range of values does not exceed 98%. The optimal values of DCP and N for achieving high probability values $\left(P_{D}>99.3\%\right)$ are the limits of N > 256 for DCP < 150. Consider Figure 9, where calculations of the probability of erroneous detection of a signaling time interval with a single-photon pulse are presented.

The figure is made for three values of the selection size (N = 256, 512, 1024) and the range of values of DCP∈{25:400}. It is apparent that the selection size N = 1024 has a significant impact on the probability at the maximum values of DCP. Thus, in the single-photon mode, the probability of erroneous detection increases sharply at DCP > 200. This is due to the fact that with the statistical accumulation of summands in Equation (4), an increase in the direct dependence of the number of DCP and the selection size causes an increase in noise signals, which are interpreted as “false positives” of single-photon avalanche photodiode. Note that the average value of DCP for the photodiodes used in QKD systems is within the range of 25 < DCP < 100. For example, the typical DCP value for id210 and id230 photodetectors is 40 and 50 Hz, respectively [18,19]. Such photodetectors are used in the Clavis^2^ and Clavis^3^ QKDS [20,21,22,23,24]. We applied the real characteristics of the id230 photodetector to our calculations (see Figure 10). The average number of photoelectrons m = 0.1 was achieved by attenuating the signal in the receiver station. The quantum efficiency of the photocathode k = 25%.

## 4. Discussion

The experimental part was strongly considered in this work. Due to the lack of a QKD system, most research groups are concerned with theoretical research. Our research team conducted theoretical research based on real experiences and found weak points by exploring real systems. By conducting experiments, we can demonstrate that this weakness can be very critical for practical application. Then, we proposed a new theoretical method to reduce the possibility of this vulnerability. The synchronization process is not part of the quantum protocol, but as shown in practice, the attacker can also access the hardware if they can access the synchronization. This can have serious consequences in real situations.

In addition, during the experiment, it was found that a new synchronization method can protect the system from quantum channel attacks. This does not represent an attack on quantum protocols but means an attack on optical communication circuits. The purpose of this attack is to destroy the key distribution.

## 5. Conclusions

Results of research show that an attack on the QKDS synchronization system can be successfully implemented. A method to counter this type of attack is presented. An important feature is that this is not a classic attack on a quantum protocol. We show that quantum key distribution systems have vulnerabilities not only in the operation of protocols. The described type of attack does not affect the cryptographic strength of the received keys, but it allows disrupting the operation of the QKDS. We are disrupting the quantum channel, but we are not interfering with the quantum protocol. Here is a simple example: if an attacker simply damages the optical cable (cuts it), the system will easily detect it; if we use our method, then the system does not detect an intruder in the quantum channel. We also propose a method that protects synchronization data from unauthorized access. The method is based on the use of sync pulses attenuated to a photon level in the process of detecting a time interval with a signal. Note that the classical attack by a compressed powerful light pulse cannot be realized, since we use an avalanche photodiode in the Geiger mode.

Synchronizing pulses are registered by single-photon avalanche photodiodes in Geiger mode. The algorithm for detecting an optical signal is described, and analytical expressions are presented for calculating probabilistic characteristics that show the undiminished dynamics of correct detection of an optical synchronizing signal. The method is simulated for optical communication systems that operate according to a two-pass scheme. The paper presents the results of experimental studies that show the vulnerability of the synchronization process in autocompensation quantum key distribution systems with phase encoding of states. An additional measure of control against unauthorized interference is the use of variable power synchronizing pulse at varying lengths of the quantum channel. Together with controlled signal attenuation, this measure will increase the security of the QKD system from unauthorized access. The results of the experiment show that the system uses pulses of the same power regardless of the length of the quantum channel. Simple calculations of sufficient synchronizing pulse power will allow the intensity of the emission source to be adjusted and pulses of calculated power to be generated depending on the length of the quantum channel.

## Figures and Tables

**Figure 1 entropy-23-00509-f001:**
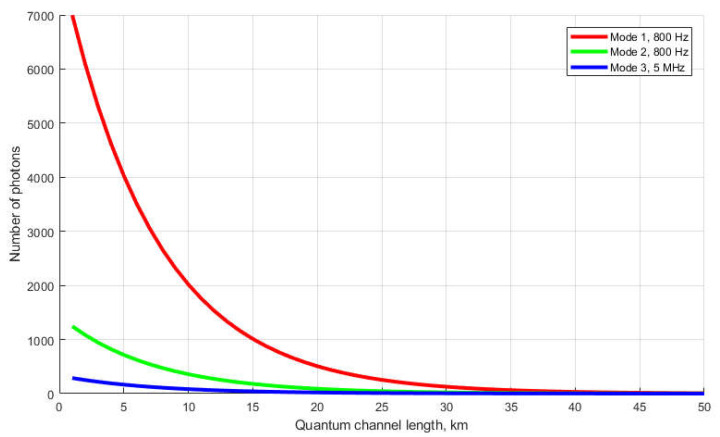
Dependence of the number of photons in a pulse on the length of the quantum channel.

**Figure 2 entropy-23-00509-f002:**
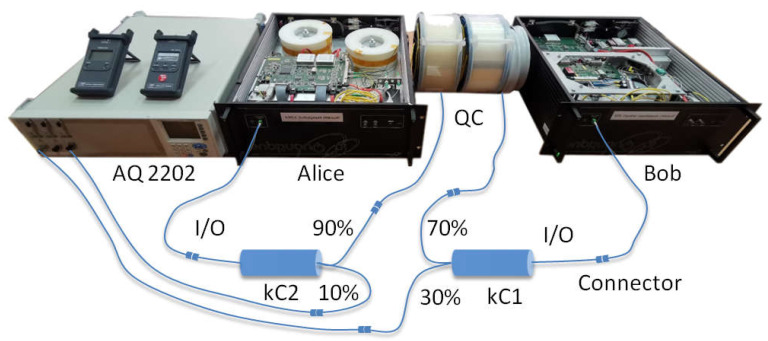
Experiment scheme. Clavis^2^ 3110 QKD system with optical power couplers (kC1, kC2). I/O is input/output.

**Figure 3 entropy-23-00509-f003:**
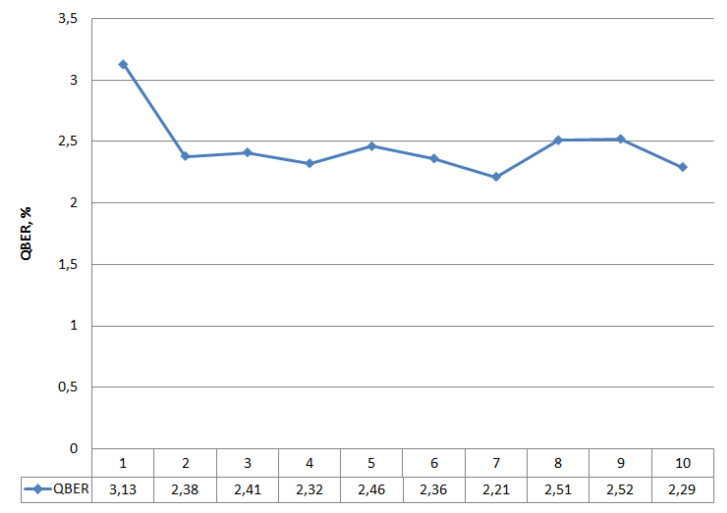
Dynamics measured by QKDS QBER software; 1–10 refer to iterations.

**Figure 4 entropy-23-00509-f004:**
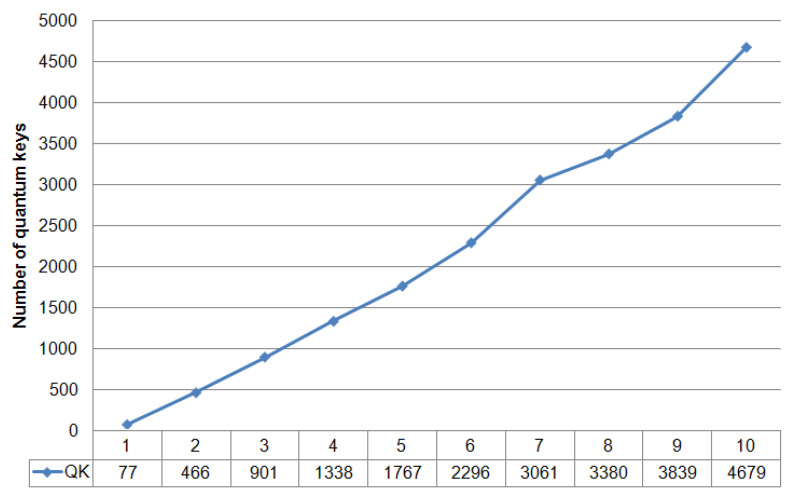
Dynamics of accumulated quantum keys. The length of each key is 512 bits; 1–10 refer to iterations.

**Figure 5 entropy-23-00509-f005:**
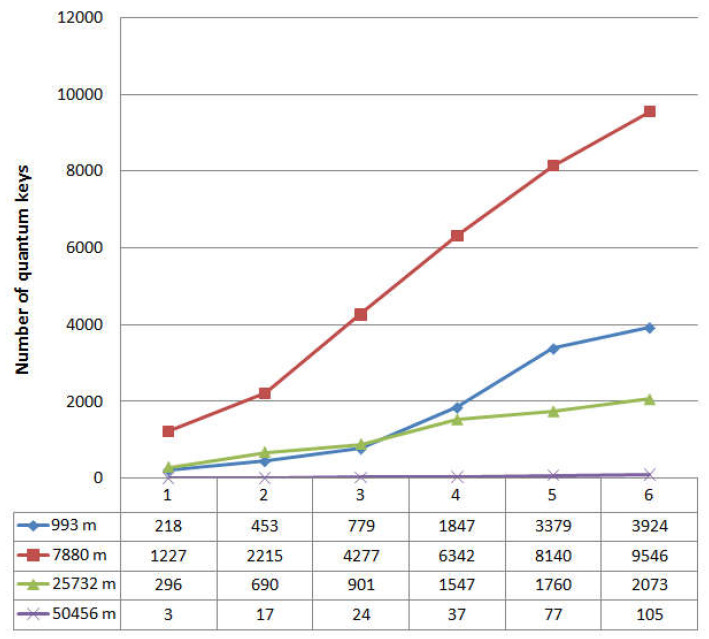
Statistical data of the BB84 quantum protocol, quantum keys; 1–6 refer to iterations.

**Figure 6 entropy-23-00509-f006:**
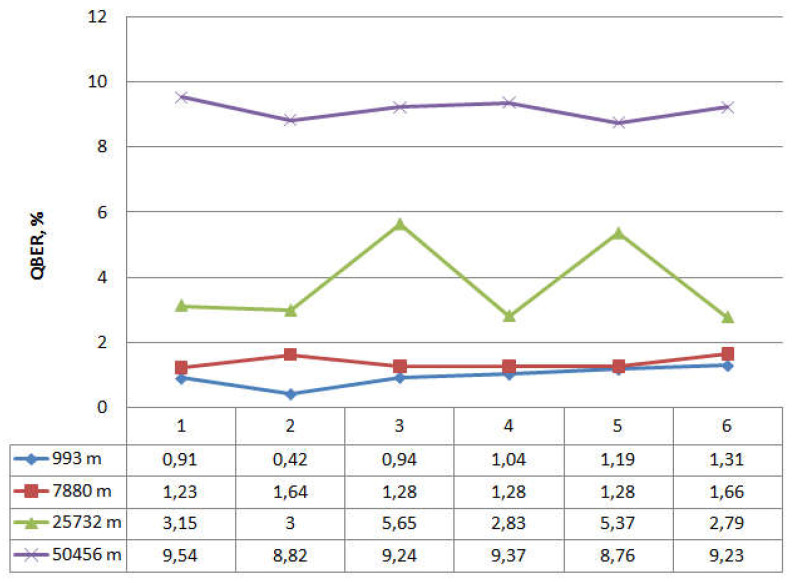
Statistical data on the operation of the quantum protocol BB84, QBER; 1–6 refer to iterations.

**Figure 7 entropy-23-00509-f007:**
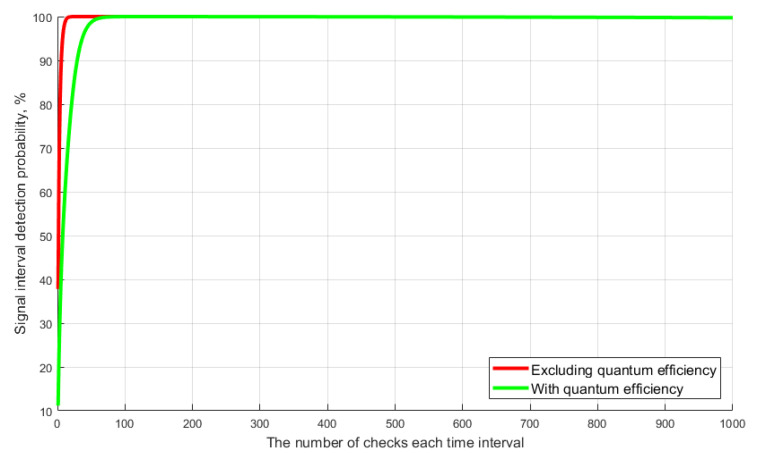
Dependence of the probability of correct detection on the selection size.

**Figure 8 entropy-23-00509-f008:**
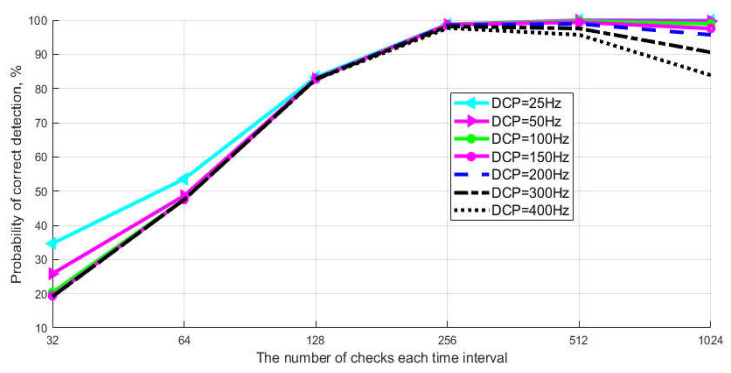
Probability of correct detection of a single-photon signal.

**Figure 9 entropy-23-00509-f009:**
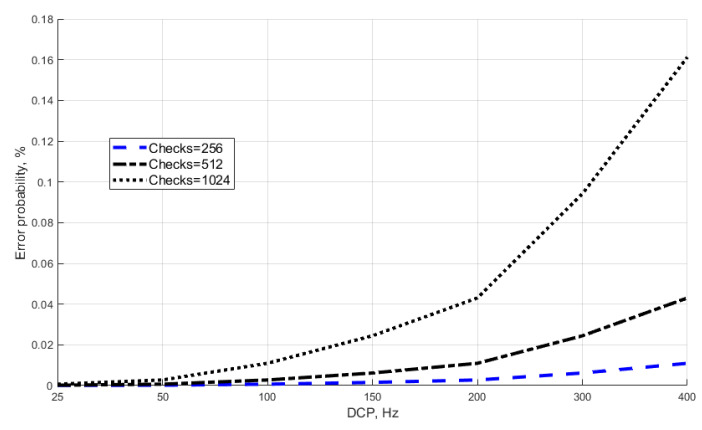
Probability of erroneous detection of a single-photon signal.

**Figure 10 entropy-23-00509-f010:**
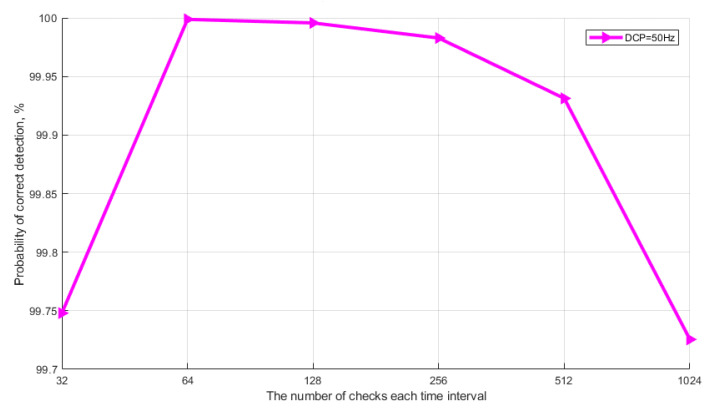
Calculating the detection probability for id230.

## Data Availability

All results and data obtained can be found in open access publications.

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
