# Peer review of "Nonclassical Attack on a Quantum Key Distribution System"

_entropy, 2021, doi:10.3390/e23050509_

Round 1

Reviewer 1 Report

The authors have analyzed the attack on the quantum key distribution system and how to protect the system from such attacks.

The subject is interesting however before giving my final decision, some critical remarks should be answered and taken into consideration for the revised version:

  • The authors introduce several concepts briefly such as quantum key distribution, attacks, synchronization process …., . The authors should write extended paragraphs on these which will increase the visibility of the paper.

The authors did not consider explicitly or at least what I have understood from the manuscript.  What about the quantum effects of the environment since the system is an open quantum system (the following references for the open system can be helpful: PRA 93, 042116 (2016) and EPJD 69, 229 (2015).

  • Is it possible by using for example squeezed light to have powerful attacks that will overcome your proposed method for reducing the vulnerability?
  • What is/are the limit(s) of your method?

Author Response

Thank you very much for your review!
We've made clarifications and changes.

"..quantum key distribution, attacks, synchronization process…"

Comment: made minor changes to the text. Inserted links detailing these processes separately. Indeed, a simple reader sometimes confuses the concepts of synchronization and protocol ..

"What about the quantum effects of the environment since the system is an open quantum system..."

Comment: Inserted links PRA 93, 042116 (2016) and EPJD 69, 229 (2015) and highlighted in the text of the article regarding the powerful impulse attack. Since we have an optical fiber system and use avalanche photodiodes (Geiger mode), such attacks will not affect our algorithm.

"What is/are the limit(s) of your method?"

Comment:  at the moment, the limit is the length of the quantum channel and the threshold value of the dark current pulses. I understand that the method requires more research and experimentation.

Reviewer 2 Report

The manuscript titled "Nonclassical attack on a quantum key distribution system" considers an unobserved intrusion into the operation of a commercial quantum key distribution system Clavis2 from ID Quantique (section 2 of the manuscript) and proposes a modification of this system, making such an unobserved intrusion impossible (section 3 of the manuscript).  

It should be noted that modern systems of quantum key distribution have various types of information exchange between the legitimate users, traditionally called Alice and Bob. The transmission of data used for establishing a secure cryptographic key is realized in a way, making unobserved intrusion impossible. In Clavis 2, this goal is reached by using very low (less than one) average number of photons in the optical link between Alice and Bob. Besides, a quantum key distribution system employs a classical channel of communication, which is open to reading for the potential adversary, and one or several protocols of technical exchange of bright pulses of light aiming at calibrating the communication line. The latter protocols do not contain any information on the key and are not intended to be secret. 

The authors of the manuscript show experimentally that insertion of optical power couplers into the fibre optical link of Clavis2, as depicted in Fig.2, allows one to monitor the calibration pulses in the fibre link and this action does not trigger the alarm of the system, whose aim is to stop the key distribution in a case of intrusion. They call this action "Nonclassical attack" and claim that this action may help the adversary to gain some knowledge on the cryptographic key by realizing another attack, briefly described in lines 131-136. Further, in section 3, they propose a modification to the system, making such an intrusion impossible, by lowering the avarage number of photons in the optical link at the calibration stage.

In my opinion, obtaining knowledge on technical parameters like properties of the calibrating pulses or optical link length, which are not intended to be secret, cannot be termed "attack". As a consequence, the remedy proposed in section 3 has no value. In summary, I do not consider this manuscript as worth publishing. In order to show that their approach is important, the authors could show how the attack, briefly mentioned in lines 131-136 can be implemented. 

Reviewer 3 Report

The paper considers an attack on a commercial QKD system.

The proposed method causes unnecessary synchronization process,

and the legitimate users of the QKD system have difficulty to detect the

presence of the attacker. The proposed method does not undermine the

security guarantee of the generated security key by the QKD system,

But the attack has practical significance of some degree.

The proposed method is described in a manner readable by the

experts in the same research field, and the experiments conducted

by the authors support the authors' claim.

I think the submitted manuscript meets the acceptance standard

by the Entropy journal.

Author Response

Thank You!

Round 2

Reviewer 1 Report

I have a comment that can be considered as a minor revision.

The sentences in 216 and2 17:

""But since the system operates on the basis of an optical fiber, such effects are described by classical functions and are reduced to the effect on the optical fiber. In most cases these influences affect the physical length of the fiber and are compensated for by checking the length in the program."

should be reformulated. The quantum fluctuations are not described by classical functions and can not be compensated.

You could mention that such quantum effects could be influencing the system but is expected their effect could be small.  

Author Response

Thank you! We have made a change to the relevant paragraph.

"We should also mention the quantum effects of the environment [17, 18]. Note that the quantum fluctuations are not described by classical functions and can not be compensated. Moreover, such quantum effects could be influencing the system but is expected their effect could be small. Of course, such effects must be taken into account and their influence on the quantum system should be investigated. There are environmental effects that can affect the physical properties of the fiber. For example, temperature that tends to change the physical length of a fiber under certain conditions but it's are compensated for by checking the length in the program."

Reviewer 2 Report

In my review report I wrote "obtaining knowledge on technical parameters like properties of the calibrating pulses or optical link length, which are not intended to be secret, cannot be termed "attack.""

The author of the manuscript acknowledges that "Indeed, we admit that it can be called something else ... for example "vulnerability of the synchronization process."" However, I do not agree to this formulation. The Merriam-Webster dictionary defines vulnerable as "open to attack or damage". The manuscript does not show that the undertaken action provides any damage to the key distribution line. 

As a consequence, I remain convinced that the results of this manuscript have no scientific value.

Author Response

Thank you again)

Yes, we may have different opinions about this procedure. Attack or vulnerability or hindrance.
I respect your opinion, but I also admit that we can talk about the same thing only in different formulations.

"The manuscript does not show that the undertaken action provides any damage to the key distribution line. "

- We are disrupting the quantum channel, but we are not interfering with the quantum protocol.
Here's a simple situation (example):
if an attacker simply damages the optical cable (cuts it), the system will easily detect it.
if we use our method, then the system does not detect an intruder in the quantum channel.